# Human Gait Modeling, Prediction and Classification for Level Walking Using Harmonic Models Derived from a Single Thigh-Mounted IMU

**DOI:** 10.3390/s22062164

**Published:** 2022-03-10

**Authors:** Nimsiri Abhayasinghe, Iain Murray

**Affiliations:** 1Department of Electrical and Electronic Engineering, Sri Lanka Institute of Information Technology, Malabe 10115, Sri Lanka; 2School of Electrical Engineering, Computing and Mathematical Sciences, Curtin University, Perth, WA 6102, Australia; i.murray@curtin.edu.au

**Keywords:** gait classification, gait prediction, harmonic modeling, human gait modeling, Inertial Measurement Units

## Abstract

The majority of human gait modeling is based on hip, foot or thigh acceleration. The regeneration accuracy of these modeling approaches is not very high. This paper presents a harmonic approach to modeling human gait during level walking based on gyroscopic signals for a single thigh-mounted Inertial Measurement Unit (IMU) and the flexion–extension derived from a single thigh-mounted IMU. The thigh angle can be modeled with five significant harmonics, with a regeneration accuracy of over 0.999 correlation and less than 0.5° RMSE per stride cycle. Comparable regeneration accuracies can be achieved with nine significant harmonics for the gyro signal. The fundamental frequency of the harmonic model can be estimated using the stride time, with an error level of 0.0479% (±0.0029%). Six commonly observed stride patterns, and harmonic models of thigh angle and gyro signal for those stride patterns, are presented in this paper. These harmonic models can be used to predict or classify the strides of walking trials, and the results are presented herein. Harmonic models may also be used for activity recognition. It has shown that human gait in level walking can be modeled with a harmonic model of thigh angle or gyro signal, using a single thigh-mounted IMU, to higher accuracies than existing techniques.

## 1. Introduction

Gait recognition is often used in infrastructure-independent navigation systems, such as inertial navigation systems. Gait analysis is performed in these systems based on trunk movement [1,2], foot movement [3,4] or thigh movement [5,6]. Some of these techniques use acceleration, whereas the others use the rotation or the orientation of the particular body section. The majority of existing studies that use inertial sensors model human gait based on the acceleration of a given section of the body. With the increasing availability of low-cost gyroscopes, the usage of these in addition to accelerometers has increased.

Although many studies exist in the literature on gait modeling, most of them present the movement pattern of a given body section for a selected group of people [7,8,9], which is of interest in clinical studies. Some others have modeled the movements of body sections based on purely theoretical models, such as mechanical models [10,11]. Detailed analyses of these works are excluded in this paper, because the scope of this work is limited to deriving a model of human gait during level walking based on empirical observations.

Ibrahim used a hip-mounted accelerometer for gait modeling, and derived harmonic models for body acceleration during different activities [2]. He used a three-axis accelerometer attached to the hip of the subjects as the source of input in his studies. A linear predictive model was used to estimate the fundamental frequency of the signal. It was discovered that there are 12 significant harmonic components in the spectrum of the acceleration signal. Harmonic models for the acceleration of the trunk during flat walking, as well as up-slope, down-slope, upstairs and downstairs walking, are presented, and the differences in the harmonic models for these activities on all three axes are discussed. The author claimed that the harmonic model derived is a good fit for the original signal, but the figures indicate that there are large deviations in the reconstructed signal from the original.

Qiuyang and Zaiyue used a spectral model to predict the thigh angle during level walking. They also used linear predictive techniques to estimate the frequencies [12]. However, in this work, they do not mention how many harmonics are required to reconstruct the original waveform accurately. The thigh angle is reconstructed using the harmonic components that are extracted and compared with the original. They reported mean square errors of 0.2 rad (11.5°) for some subjects. It can be seen from the figures that the deviation in the predicted waveform from the original is very high.

This paper presents harmonic models for thigh flexion and extension during level walking, derived from empirical data collected from a single thigh-mounted Inertial Measurement Unit (IMU) and the gyroscopic signal of a thigh-mounted IMU. The accuracy of the estimation of the fundamental frequency from the stride time was also tested against the fundamental frequency derived from Fast Fourier Transform (FFT), and is reported in this paper.

The accurate identification and classification of gait and different stride patterns plays a vital role in rehabilitation and navigation applications. The possibility of using harmonic models to predict flexion–extension angles during a walking trial, and the possibility of using harmonic models to identify the gait pattern during a walking trial, are discussed. The work presented in this paper is part of a project that is developing an indoor navigation system for vision-impaired people. Therefore, the thigh (trouser pocket) was selected as the mounting position of the IMU, based on the observations presented in [13]. It was also observed that the flexion and the extension of the thigh during walking can be estimated by performing single-dimension orientation estimation, thus reducing the computational load.

## 2. Materials and Methods

### 2.1. Experimental Setup

Thigh movement data were collected from 9 female and 10 male volunteer participants. All participants were middle-aged and are known to have no vision impairments or walking difficulties, or any other impairments. Their exact ages and body details were not recorded as this is a generic study. This study was approved by the Curtin University Ethics Committee and all volunteers gave informed consent.

A custom-made IMU [14] based on the MPU-9150 was attached to the right thigh of the subject. Figure 1 shows how the IMU was attached to the reference thigh of the subjects. Further details of the construction of the IMU and the reference details of the experiment have already been published in [14,15]. The IMU wirelessly streamed raw sensor data taken at 100 samples per second to the data-logging laptop computer. The subjects were asked to walk across a motion analysis lab (Curtin University Motion Analysis Lab, equipped with a Vicon motion capture system, as explained in [15]) with self-selected gait speed; this was taken as normal gait (approximately 2 steps per second).

### 2.2. Modeling of Thigh Angle

The flexion and extension (thigh angle) of both thighs, captured during a sample trial performed in a motion analysis lab using a Vicon motion capture system equipped with 14 semi-IR cameras during a previous study [15], are shown in Figure 2. Similar patterns were observed in all the trials conducted in the study and these have been published previously. These indicate that the two legs are in synchronization, and thus modeling one leg will help predict the movement of the other leg. Further, the same model may be applied for both legs.

Raw data collected in the experiment discussed previously were used to estimate the thigh angle using the Gyro Integration-Based Orientation Filter (GIOF) [15], which is an improved version of the algorithm discussed in [5]. GIOF derives the single-axis orientation primarily using gyro-integration, and uses accelerometer data to regularly correct the drift in the angle (when the IMU does not detect body acceleration). Then, the thigh angle was cropped from a minimum (toe-off) to a minimum excluding the first and the last strides of the walk. The rest of the analysis was performed on these cropped thigh angle waveforms.

As seen in Figure 2 and verified in previous research [16,17], the thigh angle during level walking is periodic. The most common method of modeling a periodic signal is a harmonic model: (1)$$y\left(t\right)=b+\sum_{n=1}^{N}a_{n}\cos\left(2\pi nf_{0}t+\varphi_{n}\right),\;n=1,\;2,\ldots$$
where $f_{0}$ is the fundamental frequency and *n* is the harmonic number, with $N^{th}$ harmonic as the maximum significant harmonic. $a_{n}$ and $\varphi_{n}$ are the amplitude and the initial phase of the $n^{th}$ harmonic [18]. The dc component present in the waveform is denoted by *b*.

To empirically estimate values for these unknown parameters in the harmonic model in (1), the spectrum of each trial was derived using the FFT function in Matlab. The number of frequency components in the FFT output was selected as the length of the time series sample, instead of a power of 2, due to the limited sample length. In addition to the functions available in Matlab for deriving FFT and extracting peaks, a custom-made function was used to extract the harmonics from the frequency spectrum. Figure 3 shows the frequency spectrum of a single trial, and the harmonics extracted from the spectrum (shown by circles) by the function. The function also picks up the DC component of the spectrum. It can be seen that the fundamental has the highest amplitude, which was the feature used in the custom-made Matlab function to identify the fundamental frequency, and extract the amplitudes and initial phases, of a specified number of harmonics (given as the input argument for the function). 

### 2.3. Harmonic Models for Thigh Flexion–Extension (Derived from IMU Data) and Gyro Signal

To establish harmonic models for thigh angles derived from IMU data during level walking, the thigh angle for each stride was plotted. Some stride patterns were frequently found in the set of 750 strides collected from all subjects, both male and female. Therefore, it was decided to formulate harmonic models for those frequently observed stride patters. Figure 4 depicts six such thigh angle patterns, starting with almost no oscillation after the main maximum (end of swing and heal contact) and ending with a pattern wherein the secondary peak is approximately as strong as the primary peak. The waveforms depicted in Figure 4 are samples taken from the trial, and they are normalized to have a minimum and maximum of 0 and 1. Details of how the harmonic models were derived are given later. The reasons for these stride patterns observed in the experiments are as follows.
Pattern 1—The heal contact is exactly at the end of the swing of the reference leg, and the foot is almost flat by the time of the heal contact, meaning that no oscillation of thigh is visible during the Loading Response;Pattern 2—The heal contact occurs slightly after the end of the swing of the reference leg, and there is a small angle between the foot and the ground, meaning that an angle change in the thigh is slightly visible during Loading Response;Pattern 3—The heal contact is slightly after the end of the swing of the reference leg, and there is a larger angle between the foot and the ground than in the previous case, meaning that a larger change of angle in the thigh is visible during the Loading Response;Pattern 4—The heal contact occurs after the end of the swing of the reference leg, so that the leg moves downwards before heal contact, and the foot is angled towards the ground by the time of the heal contact. Hence, an oscillation of the thigh is visible during Loading Response;Pattern 5—The heal contact occurs after the end of the swing of the reference leg, so that the leg moves downwards before heal contact and the foot is angled towards the ground by the time of the heal contact. More oscillation of the thigh is visible during Loading Response here compared to the previous case;Pattern 6—The heal contact occurs after the end of the swing of the reference leg, so that the leg moves downwards before heal contact and the foot is angled towards the ground by the time of the heal contact. A strong oscillation is visible during Loading Response, so that the secondary peak is comparable to the primary peak.

To derive harmonic models for these six patterns, sample trials with similar thigh angle patterns were collected using the following procedure.

Each stride was resampled to 2000 sample points, and normalized in time and amplitude. The stride time was normalized to 1 s, while the amplitude was normalized in such a way that the minimum of the waveform was 0 and the maximum was 1. Then, all strides with an RMSE less than 2.5% in each of the selected stride patterns (Figure 4) were extracted from the 750 strides. The stride waveforms extracted in this way were used to estimate a mean waveform representing each of the wave patterns shown in Figure 4. Harmonic models, such as those given in (1), were derived for each of these mean curves. The significant number of harmonics was decided in the fashion described in Section 3.

The spectrums of the gyro signals for each stride were also derived using FFT. In a similar way to thigh angle, the gyro signal was also reconstructed using the amplitudes and initial phases of the spectrum via (1). Here, too, the significant number of harmonics was decided as described in Section 3.

### 2.4. Predicting and Classifying Strides Using Harmonic Models

Identifying and classifying different stride patterns is important in rehabilitation and navigation applications, because prediction accuracies may be increased by using different models for different stride patterns. Therefore, the possibility of classifying strides using the harmonic models derived from IMU data was also investigated in this study. The following sections discuss how the harmonic models derived earlier can be used to predict the thigh angle waveform and classify the strides of a long walking trial.

To identify the possibility of predicting thigh angle using the harmonic model, the thigh angle of a long walk (approximately 20 strides) performed by a single subject was first reconstructed using the harmonic model extracted from the first stride. Normalized harmonic amplitudes and initial phases of the harmonics were extracted from the first stride of the trial, and these were used to recreate the thigh angle waveform for the full trial. The thigh angle was reconstructed using the timing of the stride, and then rescaled on the amplitude axis to match the peak-to-peak variation in the measured thigh angle of the stride.

Based on the observation that a person may display different stride patterns even within the same trial, the possibility of classifying strides using the harmonic model was investigated. The correlations and the RMSEs of the thigh angles for each stride to those of the thigh angles generated by the six models were computed for four different cases. The reconstructed thigh angle was scaled to match the measured thigh angle using the same method as was discussed earlier. The cases considered were walking with 3 self-selected stride rates (slow, medium and fast) on a hard floor and walking on sand (to test if this method can be used to distinguish different floor hardnesses). The variation in the stride pattern throughout the walk and in different cases can be seen by analyzing the correlation and RMSE matrices.

## 3. Results

This section presents the results obtained in each of the scenarios discussed in Section 2. For clarity, each result is presented in a separate sub section.

### 3.1. Modeling of Thigh Angle

Figure 5 shows the amplitude distribution of the first nine overtones of the frequency spectrum for all 372 trials, normalized to fundamental amplitude. It can be seen from the figure that the median amplitude becomes lower than % of the fundamental amplitude beyond the fifth overtone. (The dash–dot line close to the bottom of the plot shows 1% of the fundamental amplitude.) This 1% was selected as the cut-off because, in engineering and scientific applications, a 1% level is considered as negligible. The next level considered was 5%. However, when 5% was considered, the thigh angle waveform was observed to deviate from the original pattern, as the third harmonic is the last to be considered when the cut-off is 5%. Therefore, it can be concluded that the first five harmonics are the most significant frequency components of the thigh angle waveform during level walking.

Based on this observation, the thigh angle waveform for each trial was reconstructed using (1) with the coefficients of the first five harmonics extracted from the same trial. The original and reconstructed thigh angle waveforms of a sample trial are shown in Figure 6, and histograms for the correlations and RMSEs of the original and reconstructed waveforms of all 372 trials are shown in Figure 7. It can be seen that the majority of the correlation between the original and the reconstructed signals is greater than 0.995, while the majority of the RMSE is less than 2°, which indicates that the thigh angle waveforms can be accurately represented using the harmonic model in (1) with five harmonics. This result is much better than the correlations achieved by Ibrahim when using 12 harmonics within an accelerometer [2].

The distributions of the normalized amplitudes and initial phases of the first five harmonics of all female and male trials are shown in Figure 8 and Figure 9. The amplitudes are normalized to the fundamental amplitude. The figures indicate that the normalized amplitudes of the overtones are smaller for the males than the females in the selected sample base, but the phases of the harmonics are comparable.

The fundamental frequencies of the FFT, i.e., the stride frequencies of females, were lower than those of the males in the trials. The peak stride frequency for males was about 0.925 Hz, while for females it was close to 1 Hz. The distributions of the fundamental stride frequencies for female and male subjects are shown in Figure 10. These indicate that, in the sample, the self-selected walking speed of males was 7.5% slower. The stride frequency was also computed using the stride time and compared with the fundamental frequency extracted from the spectrum. The error in the stride frequencies estimated using stride time compared to the fundamental frequency was 0.0479% (± 0.0029%) over a total of 750 strides for all subjects, which indicates that the results derived when estimating the stride frequency using stride time are accurate.

The next stage of processing was to perform spectral analysis for each stride. Each minimum stride was extracted from the next minimum of the thigh angle waveform, and the spectral data were extracted for each stride using the same procedure to produce a harmonic model for the full trial. The extracted stride was cascaded four times before using the FFT function to derive the spectrum, in order to increase the length of the data as well as the resolution of the FFT output. In this case, the number of frequency components of the FFT output was selected as the input data length. Each stride was then reconstructed using (1), and the correlations and RMSEs were computed between the original and the reconstructed waveforms. As seen in Figure 11, the majority of correlations were greater than 0.999, and the majority of RMSE values were below 0.5°, which indicates that the model in (1) with five harmonics can be used to represent the thigh angle with high accuracy when each stride is considered.

### 3.2. Harmonic Models for Thigh Flexion–Extension (Derived from IMU data) and Gyro Signal

The harmonic models derived for the six most commonly observed thigh angle patterns are depicted in Figure 12. *n* in each of the right-hand figures indicates how many samples of similar shapes were used to derive the mean thigh angle for each pattern. Table 1 shows the normalized amplitudes and initial phases extracted for the six models. Here, too, normalization was performed with reference to the fundamental amplitude.

The histograms of correlation and RMSE between the original and reconstructed gyro signals are shown in Figure 13. It can be seen from the figure that the gyro signal can be reconstructed using the harmonic model with nine significant harmonic components to achieve better regeneration accuracies than the thigh angle waveform. The significant number of harmonics was decided upon using a similar method to that used in the case of the thigh angle.

The spectrums of the gyro signals of the aforementioned stride patterns have also been derived. The normalized amplitudes and initial phases of the first nine harmonics of the spectrum of the gyroscopic signals of the six most commonly observed stride patterns are shown in Figure 14, and the coefficients are tabulated in Table 2.

### 3.3. Predicting Thigh Angle Using the Harmonic Model

In order to predict the stride pattern, each stride was regenerated using the harmonic model extracted from the first stride of the trial. For each stride, the fundamental frequency was estimated using the stride time, and the maximum thigh angle was used as the fundamental amplitude. The original thigh angle waveform, and the thigh angle reconstructed using the harmonic model in (1) with five significant harmonics, are shown in Figure 15. The trial was reconstructed with an RMSE of 1.33° and a correlation of 0.997. It can be seen in the figure that the reconstructed waveform deviates from the original for some strides. This is because a person may have different stride patterns, as discussed in Section 3.2. This observation constitutes the basis of the work presented in the following discussion.

### 3.4. Classifying Strides Using the Harmonic Model

In order to classify each stride of a long walk, the correlations of each stride to the base models discussed in Section 3.2 were estimated. Figure 16 depicts the correlation matrices as color maps. Each matrix shows the correlation of each stride in a given trial to the pattern reconstructed using each model. Each column of a matrix represents a single stride, and a row represents a particular model.

It can be seen in the figure that all the strides achieved a better correlation with Model 1 for slow walking, whereas for medium walking, most of the strides showed a closer correlation to Model 3, and other strides matched with the patterns of Models 1, 4 or 6. For fast walking, the stride pattern was that of Model 6, while on sand it became Model 1, except for a few strides, which were Model 3. According to the observations made during the trial on sand, the stride pattern became that of Model 3 when the sand was hard, and Model 1 when the sand was loose. These observations indicate that the harmonic models of thigh angle can be used to identify the variations in stride during a single walking trial.

The classifications of strides using the correlation of the original stride waveform to the reconstructed waveform may be verified using the RMSE between the stride waveform and the model. Figure 17 shows the RMSE matrices as color maps for aforementioned scenarios. It is clear that the RMSE matrices also follow the same patterns as the correlation matrices. The closer the correlation of the thigh angle waveform of a particular stride and the waveform generated by a particular model to 1, the closer the shape of the stride is to the shape of the model. In the case of RMSE, the smallest value indicates the best model. Hence, the best model that most closely represents the shape of a particular stride is the one that shows the highest correlation and the smallest RMSE.

The same method of classification can be performed using the gyro signal of each stride. As seen in Figure 14, the toe-off point is the point at which the gyro signal is crossed at zero with a positive gradient. The classification performed with gyro signal also gives similar results, as shown in Figure 18. Tis figure shows that the stride pattern here is closer to Model 1 than the others. The full set looks similar to the one for the thigh angle. The full set is not included in Figure 18 to reduce the complexity. This indicates that classification and reconstruction can be performed using the gyro signal.

## 4. Discussion

Harmonic models for the thigh angle and gyroscopic signal have been presented in this paper. By analyzing the spectrums of the thigh angle of 372 level walking trials performed by 19 subjects, both male and female, it was observed that the first five harmonics are significant. A correlation greater than 99.5% and an RMSE less than 2° between the original thigh angle waveform and the one reconstructed from the derived harmonic model indicates that the thigh angle waveform can be effectively reconstructed with a harmonic model using the first five harmonics. For the gyro signal, the number of significant harmonic components is nine. The thigh angle and the gyro signal for each stride were reconstructed with the harmonic models, achieving correlations greater than 99.9% and 99.99% and RMSE values less than 1° and 5% respectively, which is a better fit than that of the full trial.

Harmonic models of thigh angle and gyro signal for six commonly observed stride patterns are also presented in this paper. The models were used to classify strides during a trial with a long flat walk (about 20 strides). It was shown that the correlation and the RMSE between the original and the model-generated signals may be used together to identify the stride patterns. Correlations above 99% and RMSE values less than 2° were observed for the best matching model. The best correlation and RMSE values were used to indicate the model that most closely matched the particular stride in the classification process. The thigh angle waveform of a long level walk was reconstructed using the harmonic model extracted from the first stride with five harmonics, achieving a correlation of 99.7% and an RMSE of 1.3°.

All the correlations and RMSE figures reported in this study are far better than the correlation and error figures achieved in [2,12]. Chhoeum et al. reported 75% and 97% correlation coefficients [19], but they used specially made textile capacitive sensors in the footwear in their study. However, the design costs and complexity of such a system are higher. Many other references (e.g., [20]) do not clearly present the correlations between the original and the regenerated waveforms.

Harmonic models for common stride patterns may be derived from a set of samples for the purpose of classifying a selected group of subjects. These groups may be people with no impairment or disability, patients with a certain lower limb disability, or patients who have undergone lower limb surgery. These models may then be used to classify or identify stride patterns in order to identify certain lower limb disabilities, or for pre- and post-surgery movement analysis; however, they may require higher sampling rates and sensors with higher accuracy in order to identify fine details of the gait pattern.

Furthermore, these harmonic models may be used to identify a person’s stride patterns for navigation applications. Harmonic models may be extracted for a given subject to identify different stride patterns that a particular person has, in the context of different terrains, activities or footwear. Such models may be used to improve pedestrian dead reconning accuracy, as well as in activity recognition. Figure 19 shows some exemplar stride patterns during upwards and downwards walking on stairs and ramps. It can be seen that these stride patterns are quite different from the patterns displayed during level walking, which results may be used in activity recognition.

It was also shown that stride modeling and classification can be performed using the gyro signals and the harmonic models derived for selected thigh angle patterns. However, in this case, the first nine harmonics became significant, which will increase the computational demands.

Thigh angle waveform patterns similar to those observed for non-vision-impaired subjects were observed in vision-impaired subjects. Therefore, the same techniques may be applicable to vision-impaired subjects too.

In conclusion, the thigh angle estimated using a single IMU can be reconstructed using a harmonic model with five harmonics and a gyro signal with nine harmonics with great accuracy. These harmonic models may be used for many different navigation and rehabilitation applications.

## Figures and Tables

**Figure 1 sensors-22-02164-f001:**
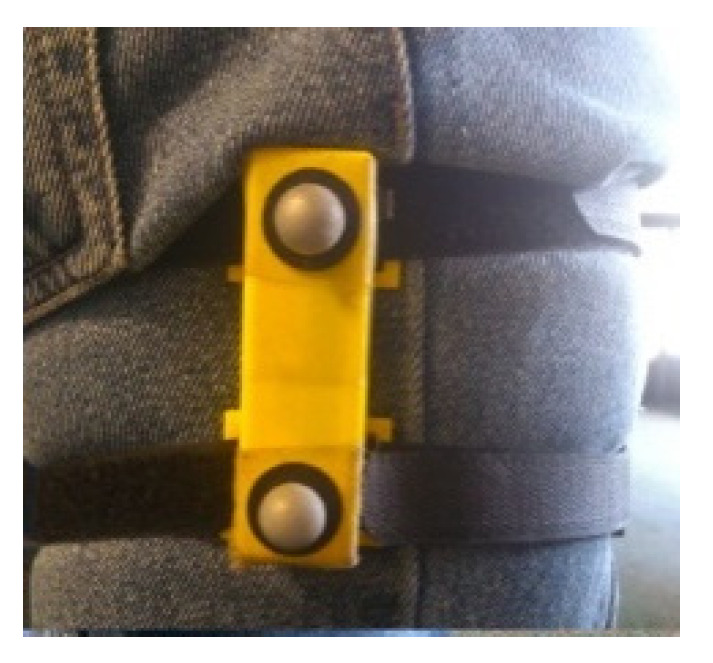
Attachment of IMU to the reference thigh of the subjects.

**Figure 2 sensors-22-02164-f002:**
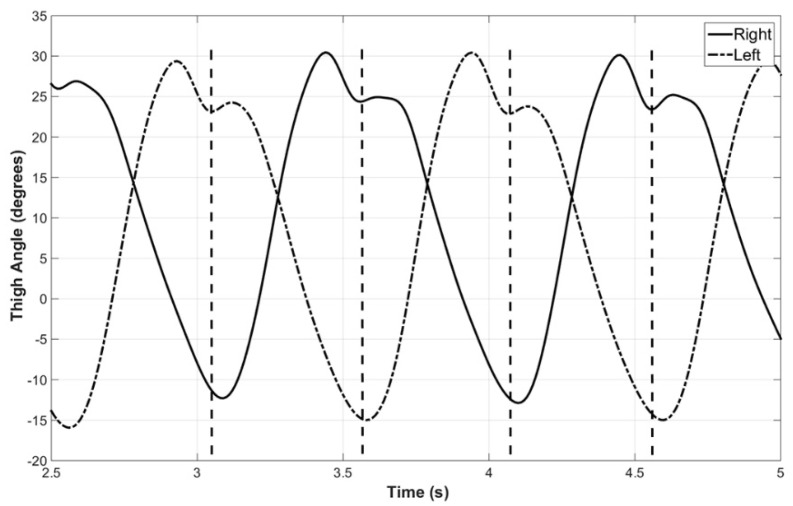
Thigh angles of left and right legs. This indicates that the toe-off of one leg is synchronized with the initial contact of the other leg, as shown by vertical dashed lines [15].

**Figure 3 sensors-22-02164-f003:**
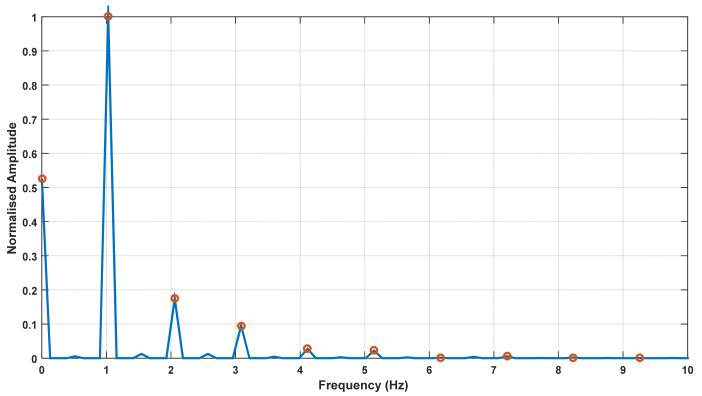
Frequency spectrum of a sample trial and harmonic components extracted from the function (shown by circles).

**Figure 4 sensors-22-02164-f004:**
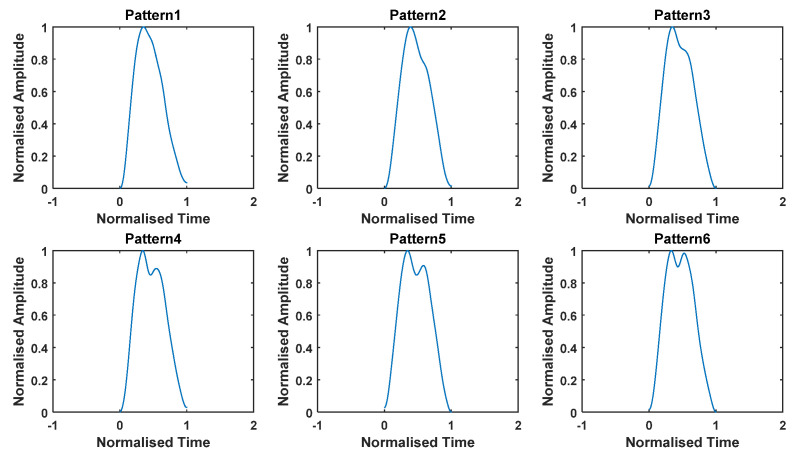
Six common thigh angle patterns observed in all trials.

**Figure 5 sensors-22-02164-f005:**
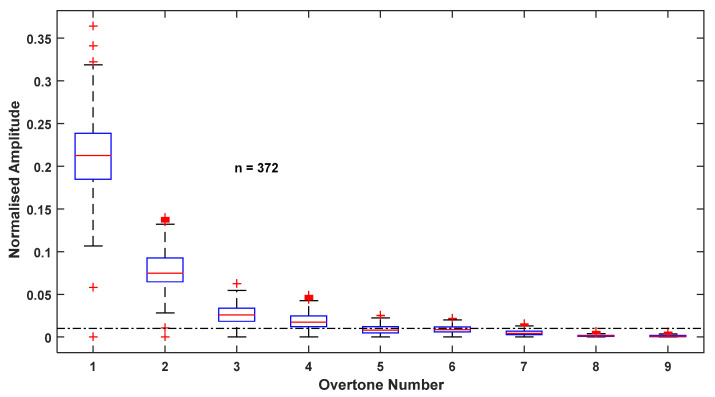
Amplitude distribution of first nine overtones of all trials normalized to the fundamental amplitude.

**Figure 6 sensors-22-02164-f006:**
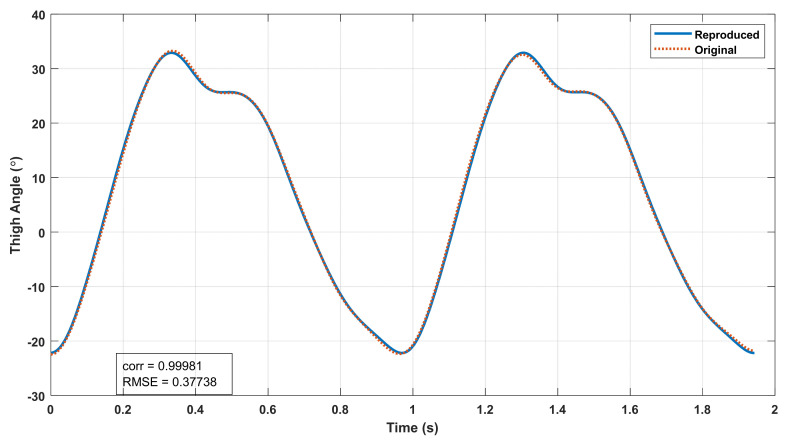
Original and reconstructed thigh angle waveforms of a sample trial.

**Figure 7 sensors-22-02164-f007:**
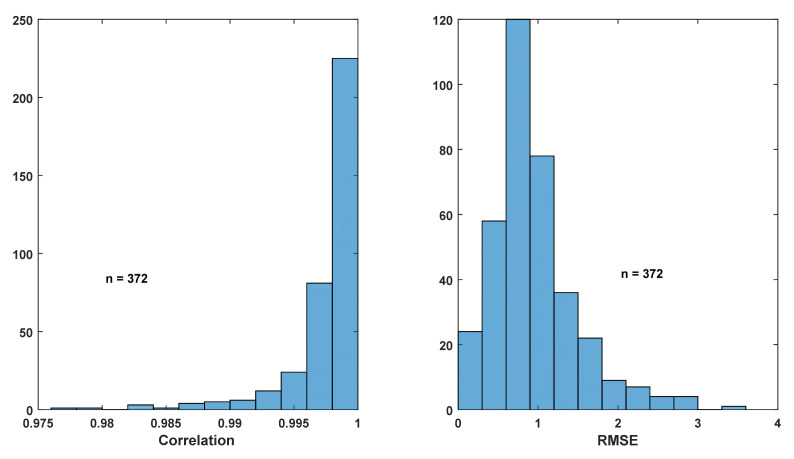
Histograms of correlation and RMSE between original and reconstructed thigh angle waveforms using five harmonics.

**Figure 8 sensors-22-02164-f008:**
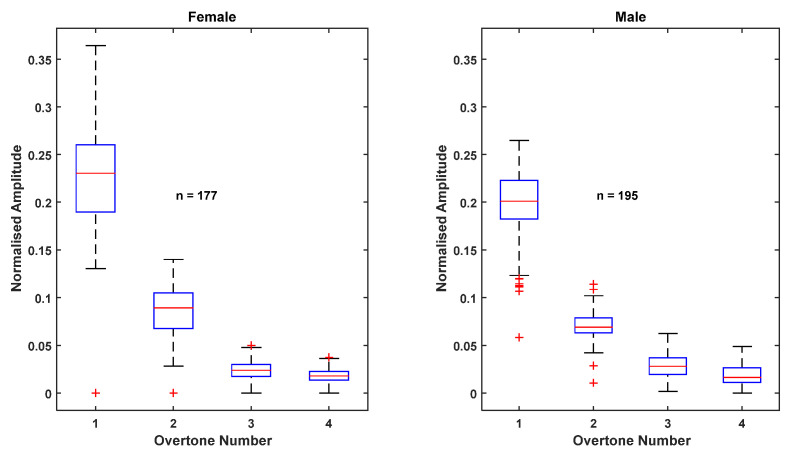
Distributions of normalized harmonic amplitudes of first four overtones for all female and male trials.

**Figure 9 sensors-22-02164-f009:**
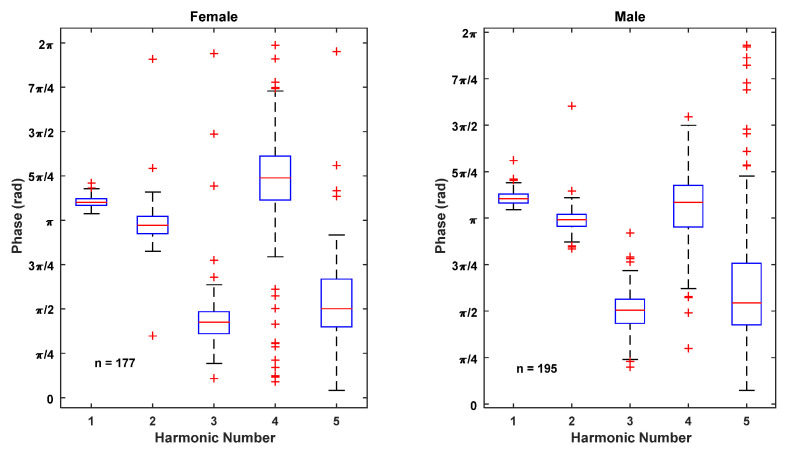
Distributions of initial phases of first five harmonics for all female and male trials.

**Figure 10 sensors-22-02164-f010:**
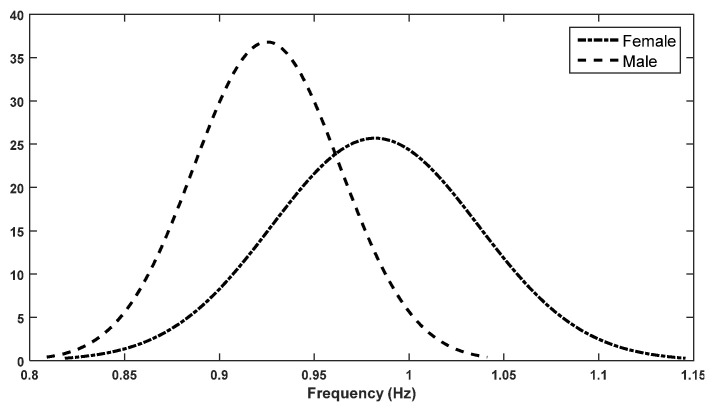
Distribution of fundamental frequencies (stride frequencies) of female and male subjects.

**Figure 11 sensors-22-02164-f011:**
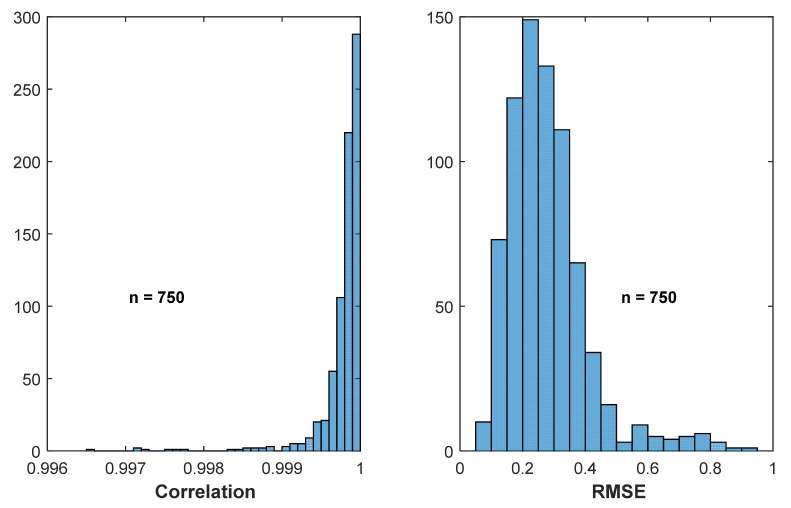
Correlation and RMSE between original and reconstructed thigh angles for each stride.

**Figure 12 sensors-22-02164-f012:**
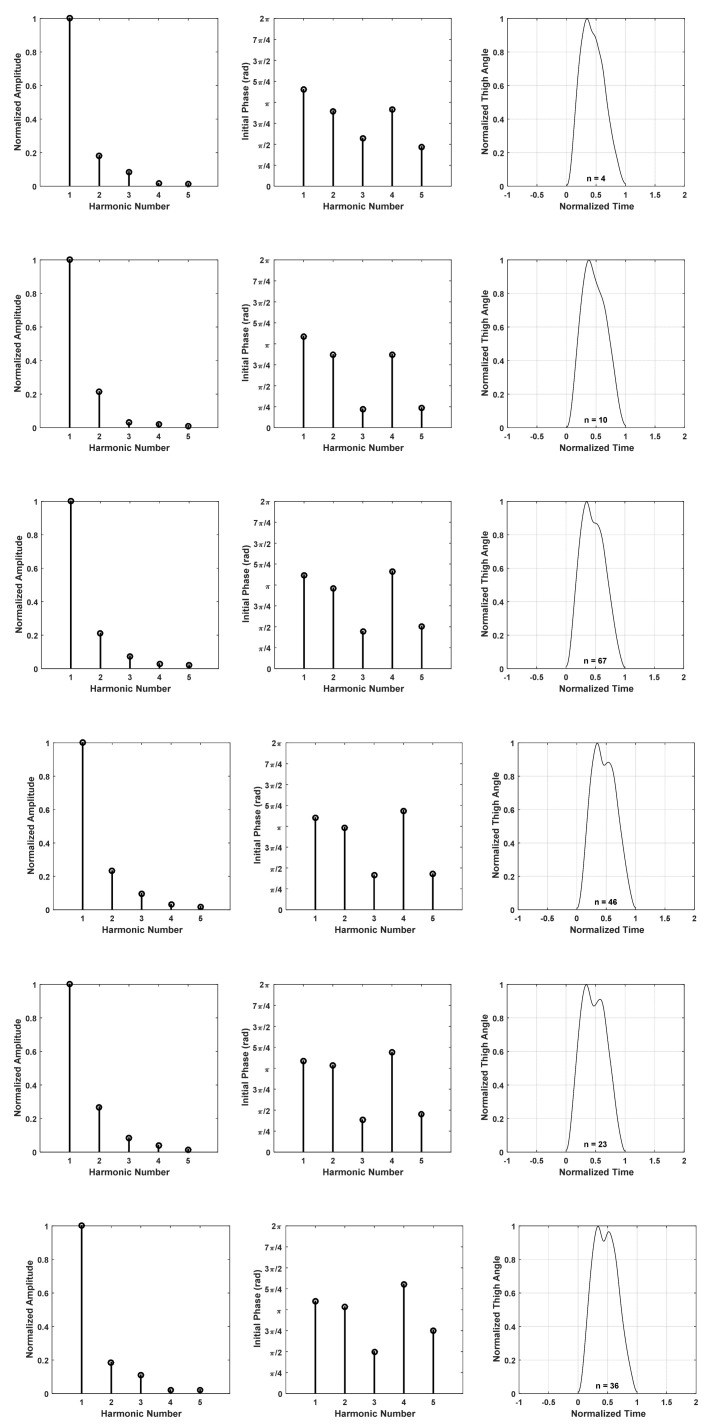
Harmonic models for six commonly observed thigh angle patterns.

**Figure 13 sensors-22-02164-f013:**
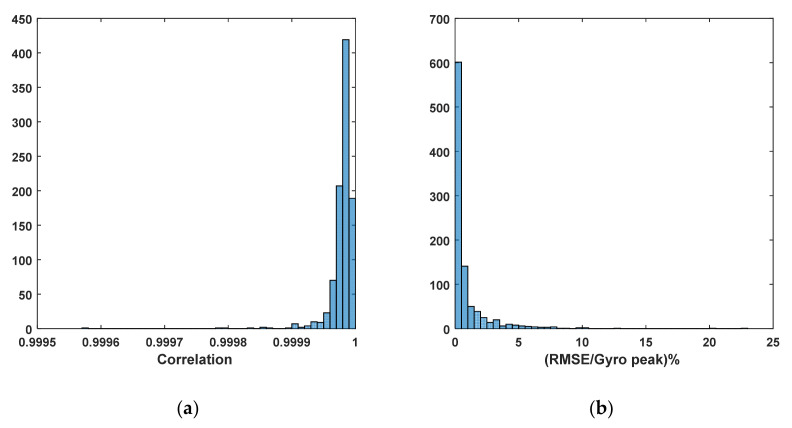
Correlation (**a**) and RMSE (**b**) between original and reconstructed gyro signal for each stride.

**Figure 14 sensors-22-02164-f014:**
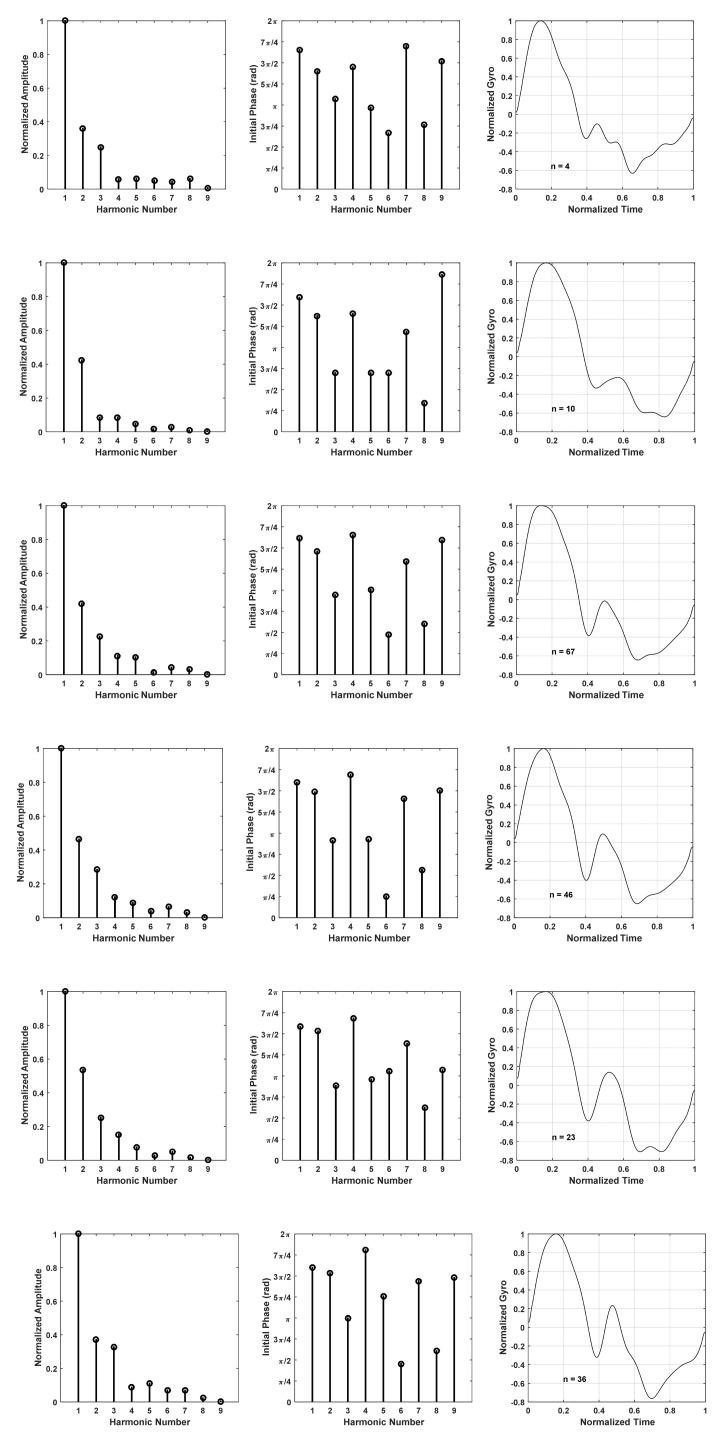
Harmonic models of gyro signal for six commonly observed thigh angle patterns.

**Figure 15 sensors-22-02164-f015:**
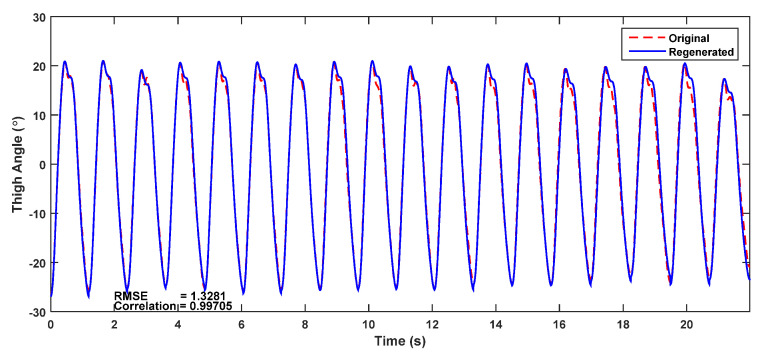
Thigh angle waveform reconstructed using a harmonic model with five harmonics and the original thigh angle waveform for a long walk.

**Figure 16 sensors-22-02164-f016:**
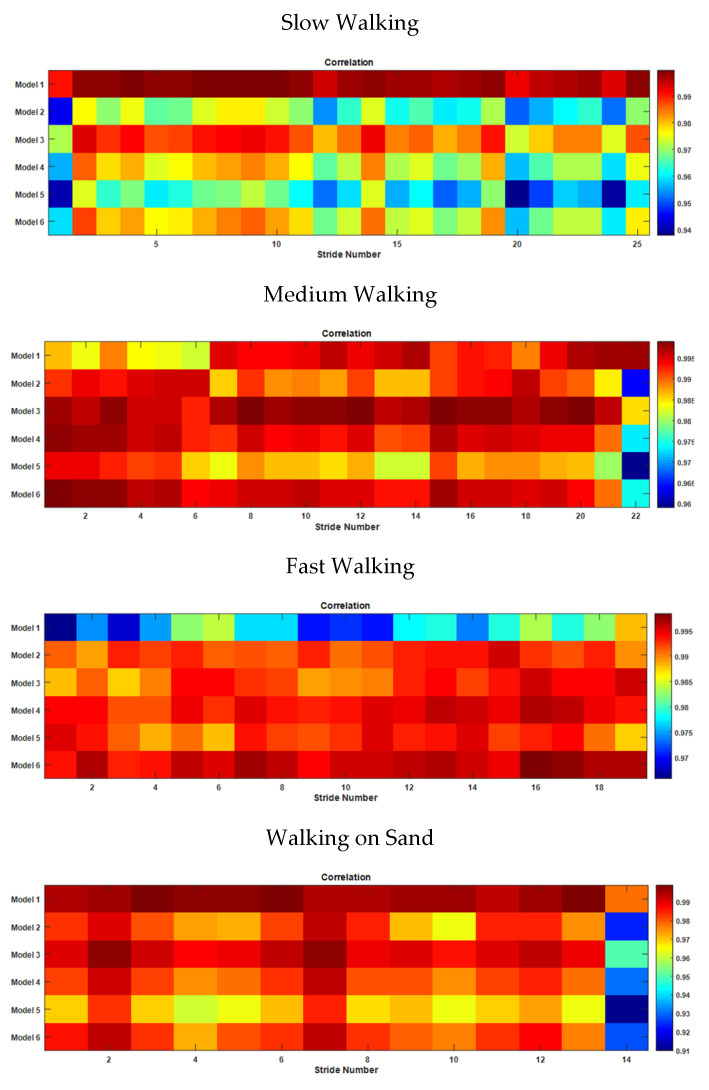
Correlation of each stride with model-generated waveform for different cases.

**Figure 17 sensors-22-02164-f017:**
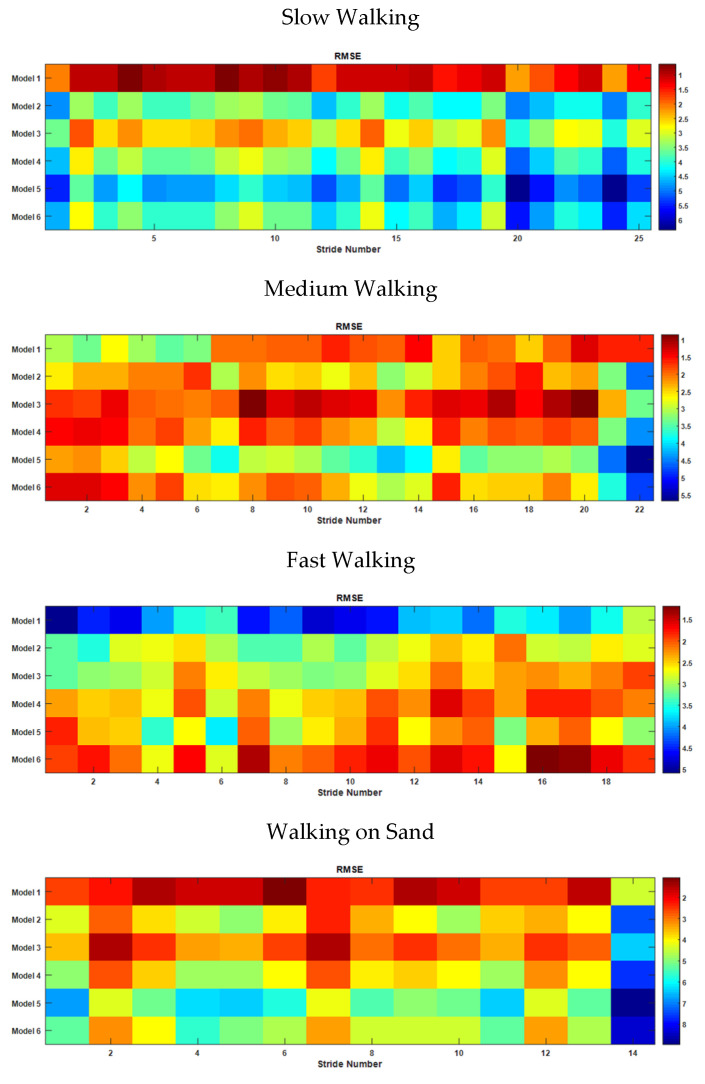
RMSE of each stride with the model-generated waveform for different cases.

**Figure 18 sensors-22-02164-f018:**
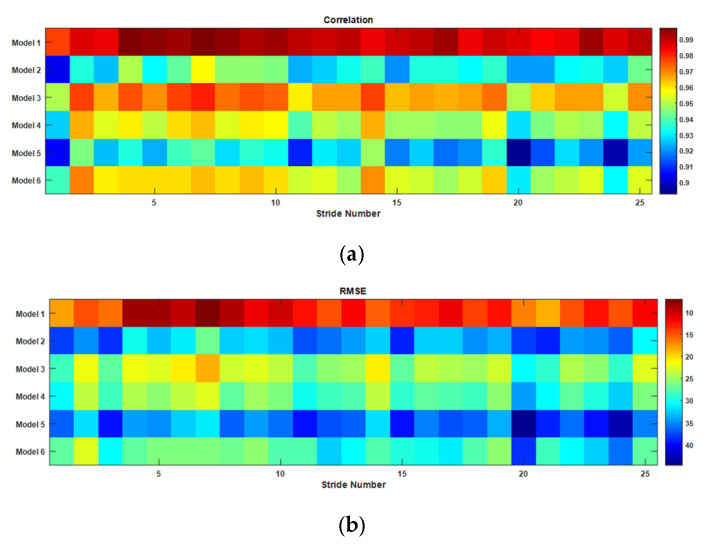
The correlation (**a**) and RMSE (**b**) of the gyro signal of each stride to the model-generated waveform for slow walking.

**Figure 19 sensors-22-02164-f019:**
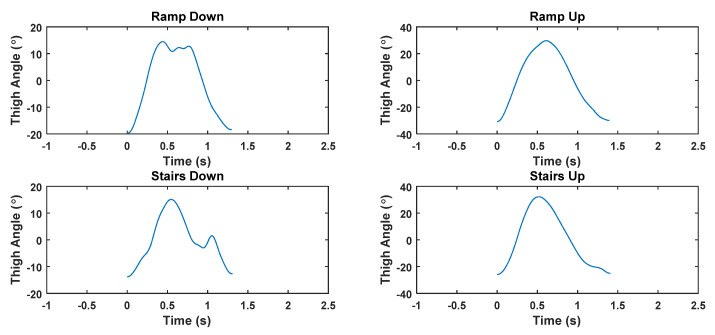
Thigh angle patterns for different activities.

**Table 1 sensors-22-02164-t001:** Coefficients of harmonic models of the six most common thigh angle patterns.

Model	1st Harmonic		2nd Harmonic		3rd Harmonic		4th Harmonic		5th Harmonic	
	*a_n_*	*ϕ_n_* (rad)	*a_n_*	*ϕ_n_* (rad)	*a_n_*	*ϕ_n_* (rad)	*a_n_*	*ϕ_n_* (rad)	*a_n_*	*ϕ_n_* (rad)
1	1	3.6133	0.18150	2.8125	0.08525	1.7860	0.015085	2.8816	0.013952	1.4683
2	1	3.4174	0.21348	2.7200	0.03001	0.6868	0.022038	2.7407	0.010615	0.7321
3	1	3.5088	0.20959	3.0176	0.07352	1.3860	0.028212	3.6534	0.019884	1.5846
4	1	3.4504	0.23148	3.0875	0.09581	1.2984	0.030199	3.727	0.018138	1.3387
5	1	3.4057	0.26781	3.2431	0.08322	1.1994	0.038494	3.7332	0.014827	1.4261
6	1	3.4515	0.18452	3.2409	0.10950	1.5581	0.021041	4.0952	0.022180	2.3650

**Table 2 sensors-22-02164-t002:** Coefficients of harmonic models of gyro signals of the six most common thigh angle patterns.

Model	1st Harmonic		2nd Harmonic		3rd Harmonic		4th Harmonic		5th Harmonic	
	*a_n_*	*ϕ_n_* (rad)	*a_n_*	*ϕ_n_* (rad)	*a_n_*	*ϕ_n_* (rad)	*a_n_*	*ϕ_n_* (rad)	*a_n_*	*ϕ_n_* (rad)
1	1	5.19	0.36027	4.402	0.2489	3.3649	0.059486	4.5546	0.063368	3.0386
2	1	4.996	0.42343	4.3092	0.084499	2.1904	0.085638	4.3984	0.047704	2.1871
3	1	5.0759	0.42098	4.58	0.22486	2.9632	0.11145	5.1886	0.10383	3.1609
4	1	5.0243	0.46313	4.666	0.28408	2.8688	0.12266	5.3146	0.087658	2.9082
5	1	4.9736	0.53663	4.8087	0.25256	2.7775	0.15274	5.2844	0.077172	3.0107
6	1	5.0247	0.36973	4.8195	0.32584	3.1308	0.086019	5.6766	0.10968	3.9544
**Model**	**6th Harmonic**		**7th Harmonic**		**8th Harmonic**		**9th Harmonic**			
	** *a_n_* **	** *ϕ_n_* ** **(rad)**	** *a_n_* **	** *ϕ_n_* ** **(rad)**	** *a_n_* **	** *ϕ_n_* ** **(rad)**	** *a_n_* **	** *ϕ_n_* ** **(rad)**		
1	0.051567	2.0871	0.043595	5.3394	0.06063	2.397	0.006469	4.7733		
2	0.01632	2.187	0.028081	3.7139	0.008522	1.0669	0.00339	5.8446		
3	0.014819	1.4992	0.042851	4.2084	0.031311	1.8894	0.002707	4.9969		
4	0.039558	0.79304	0.064016	4.4247	0.030717	1.7806	0.002826	4.7229		
5	0.029039	3.3201	0.049497	4.3447	0.015691	1.9591	0.000666	3.3624		
6	0.069381	1.4251	0.067827	4.5243	0.022453	1.9092	0.000749	4.6469		

## Data Availability

The data used in this study were collected for the study and are not available for public use.

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
