# Peer review of "Human Gait Modeling, Prediction and Classification for Level Walking Using Harmonic Models Derived from a Single Thigh-Mounted IMU"

_sensors, 2022, doi:10.3390/s22062164_

Round 1
Reviewer 1 Report
Overall, this paper needs to be extensively edited for English language usage.
The very last paragraph of the introduction is probably not needed. Section 5 – Classification of Strides Using Harmonic Models is very interesting. However, this is not really alluded to in the introduction. Lines 243-248 Should be moved to the introduction to make this more clear.
Experimental Setup – shouldn’t this section be titled “Methods”?
What are the ages of the participants, height, weight, BMI? Was this study approved by an institutional review board or ethics committee? Did the participants undergo informed consent?
The authors mention a ‘motion analysis lab’. What equipment was used? Is this a camera system? If so, how many cameras, where were markers placed, etc. The authors need to describe this better.
Figure 1: Is this an example of one subject or some sort of average? If this is one subject, was this true for all of them?
Line 94: Briefly explain GIOF.
Lines 122-127: How was the fifth amplitude chosen as your cutoff? Why not the 4th or 3rd? Could you apply a statistical methodology to justify the cutoff?
Line 156: Was this difference statistically significant?
Figure 10 is very interesting. However, the y axis shows ‘normalized’ amplitude. How were the data normalized? Or is this some sort of average?
Figure 11: Same question, how were the data normalized?
Lines 305-307: This is a good explanation of how the cutoff was determined and should be included earlier.
The first two paragraphs of the discussion are just summaries of the results. There isn’t any comparison to previous studies, until the very end of paragraph two. There is also not much explanation of what these results mean. I would suggest deleting this or placing the results in the context of the body of work that has been previously published. The authors should expand their literature search.
Figure 13: Is this data from the current study? It’s not clear why this is included here.
Reviewer 2 Report
- I would suggest the authors adding schematic diagram of the data acquisition system in sectionâ…¡.
- Line110/page3.A: please, specify the custom-made function.
- Why the authors choose 1% of the fundamental amplitude as the threshold? Whether other values can also be used as discriminant thresholds.
- It is suggested that the authors change some pictures in the article to improve the clarity. Figure 2,4,10,13,14,18.
- It is not specified in Fig. 17 if the classification and reconstruction can be performed using gyro signa in different walking cases(medium , fast and walking on sand).
- There are several typos and syntax errors throughout the document, showcasing that no formal proofreading has been performed. See for example line37-38/page1,line88/page2,line104/page3.
Reviewer 3 Report
The paper presents harmonic modelling of human gait during walking based on gyroscopic signal of a single thigh mounted inertial measurement unit. The paper is interesting and generally well written. However, three issues need to be solved:
- Lines 70-74: instead of Arabic numerals to denote sections, the author used Roman numerals. See also line 93 and 260.
- Section 2 is very short to be a section. It needs either to be enlarged (by giving some details) or to be included as a subsection of Section 3;
- Gait prediction is only mentioned (see the paper title) but not discussed in the paper. How exactly it can be done?
